# High-energy and low-cost membrane-free chlorine flow battery

Singyuk Hou [1,5], Long Chen [1,2,5✉], Xiulin Fan[1,5], Xiaotong Fan[3], Xiao Ji [1], Boyu Wang[1], Chunyu Cui[1], Ji Chen[1], Chongyin Yang [1], Wei Wang [4], Chunzhong Li [2] & Chunsheng Wang [1✉]

Grid-scale energy storage is essential for reliable electricity transmission and renewable energy integration. Redox flow batteries (RFB) provide affordable and scalable solutions for stationary energy storage. However, most of the current RFB chemistries are based on expensive transition metal ions or synthetic organics. Here, we report a reversible chlorine redox flow battery starting from the electrolysis of aqueous NaCl electrolyte and the as-produced $Cl_2$ is extracted and stored in the carbon tetrachloride ($CCl_4$) or mineral spirit flow. The immiscibility between the $CCl_4$ or mineral spirit and NaCl electrolyte enables a membrane-free design with an energy efficiency of >91% at $10 \, mA/cm^2$ and an energy density of 125.7 Wh/L. The chlorine flow battery can meet the stringent price and reliability target for stationary energy storage with the inherently low-cost active materials (~\$5/kWh) and the highly reversible $Cl_2/Cl^-$ redox reaction.

[1] Department of Chemical and Biomolecular Engineering, University of Maryland, College Park, MD, USA. [2] Department of Chemical Engineering, East China University of Science and Technology, Shanghai, China. [3] School of Materials Science and Engineering, East China University of Science and Technology, Shanghai, China. [4] Energy & Environment Directorate, Pacific Northwest National Laboratory, Richland, WA, USA. [5] These authors contributed equally: Singyuk Hou, Long Chen, Xiulin Fan. ✉email: longchen@ecust.edu.cn; cswang@umd.edu

ntegrating renewable energy, such as solar and wind power, is essential to reducing carbon emissions for sustainable development. However, large-scale utilization is hindered by the intermittence and uneven distribution of these power sources[1–3]. Implementation of grid-scale energy storage is essential to mitigate the mismatch between electricity production and consumption[4]. Different technologies are developed for this purpose, including supercapacitors, sodium–sulfur batteries, pump hydro, flywheels, and superconducting magnetic energy[5]. Redox flow battery (RFB) is considered one of the most attractive energy storage systems for large-scale applications due to the lower capital cost, higher energy conversion efficiency, and facile modularity[6,7]. The cores of flow cells are the circulating electrolytes that carry the redox-active materials for energy storage and release.

Currently, the all-vanadium RFB is the most researched and developed RFB chemistry; however, the market adoption of this system has been hampered by high-cost chemicals (material cost close to 60% of the overall system cost[8] and low energy density. Although aqueous soluble organic redox species offer a potential option for low-cost materials[9–15], the synthetic processes required to customize the molecular structure for high solubility and optimal potential will again limit the material cost and availability[6,16–18]. Also, they rely on the costly ion-permeable membranes to reduce cross-over, further increasing capital and maintenance costs[19].

Recently, polymer redox couples were developed to circumvent ion-permeable membranes[20], and the semi-solid Li-ion (suspensions of Li-ion battery active materials in nonaqueous electrolytes) systems have been explored for higher energy density and efficiency. However, high viscosity, lower peak power operation time, and high material cost emerged with these systems[4,21,22].

To meet the needs of RFB chemistries with the naturally abundant and low-cost redox-active materials, we report a new RFB system that capitalizes the electrolysis of saltwater or aqueous NaCl electrolyte using the $Cl_2/Cl^-$ redox couple as the active material for the positive electrode. The $Cl_2/Cl^-$ has a theoretical capacity of 755 mAh/g, more than two times that of vanadium oxides ($VO_2^+/VO^{2+}$, 226 mAh/g) used in current RFBs. $Cl_2/Cl^-$ redox chemistry is a fast single-electron transferred reaction with an activation energy of 35.5 kJ/mol[23,24], which is comparable to or even smaller than that of $VO_2^+/VO^{2+}$[25], thus is suitable for high power applications. In addition, sodium chloride is one of the cheapest commodities available due to the abundant source in seawater and large-scale production (~$40 per metric ton)[26,27]. These features enable $Cl_2/Cl^-$ redox reaction to be a promising candidate for RFB.

Rarely heard in the battery history is that $Cl_2/Cl^-$ redox couple was used in the RFB to power the first fully controlled airship La France in 1884[28]. The $Cl_2/Cl^-$ based batteries are often typified by low Coulombic efficiency (CE) of 40–70%[29–33] due to $Cl_2$ dissolution in the electrolytes and large voltage hysteresis (0.7 V at 32 mA/cm$^2$) due to non-wettability between electrolytes and electrodes[34,35], which limits the energy efficiency to around 60%. Graphite was reported as chlorine storage host via intercalation[36]. However, the instability of $Cl_2$ intercalated graphite at room temperature results in low storage capacity (35–40 mAh/g) and limited cycle life. After that, no other materials with appropriate stability, storage capacity, and reaction kinetics have been reported to enable reversible $Cl_2$ electrochemical reaction.

Our objective is to develop a new RFB with the highly reversible $Cl_2/Cl^-$ redox species through electrolyzing the saturated NaCl aqueous electrolyte ($NaCl/H_2O$) and storing the as-produced $Cl_2$ in water-immiscible organic phases such as carbon tetrachloride ($CCl_4$) or mineral spirits. These organic phases provide several desirable properties: (1) $Cl_2$ in $CCl_4$ ($Cl_2$-$CCl_4$) delivers a volumetric capacity of 97 Ah/L due to high solubility of $Cl_2$ in $CCl_4$ (0.184 mole/mole $CCl_4$[37], which is a 2 to 4 times improvement over the current vanadium-based catholyte (22.6–43.1 Ah/L[38]; (2) The $Cl_2$-$CCl_4$ is immiscible to $NaCl/H_2O$, thus requires no membrane to prevent cross-over, further reducing costs; (3) The $Cl_2$-$CCl_4$ has low and constant viscosity of 0.819 mPa.s, in contrast to high and varying viscosity of aqueous vanadium-based catholyte (1.4–3.2 mPa.s[39], thus is easy to flow; (4) $Cl_2$-$CCl_4$ can wet carbon porous electrodes easily, which significantly enhances the surface area for $Cl_2$ storage and reaction; (5) $Cl_2$ has high diffusivity in $CCl_4$, minimizing energy dissipation for mass transport.

## Results

**Storage and electrochemical performance of $Cl_2$-$CCl_4$.** The $Cl_2/Cl^-$ redox reaction in $NaCl/H_2O$ was evaluated in a concentric cell with $RuO_2$-$TiO_2$ coated porous carbon ($RuO_2$-$TiO_2$@C) as a working electrode, activated carbon as a counter electrode (Fig. S1), and Ag/AgCl as the reference electrode (Fig. 1A). The $RuO_2$-$TiO_2$ catalysts on porous carbon (Figs. S2, S3) are used to promote the oxidation kinetics of chloride[40–43] (Fig. S4). $CCl_4$ was pumped through the working electrode, and the $NaCl/H_2O$ through the interstitial space between the working and counter electrodes to ensure adequate $Cl^-$ supply. While $CCl_4$ and $NaCl/H_2O$ entered the $RuO_2$-$TiO_2$@C electrode as separate flows, they both wet the carbon electrode, demonstrated by <90° contact angles (CAs) on a graphite plate electrode (Fig. 1B, C). And the two liquids take up 66.2% and 33.8% of the void volume in the $RuO_2$-$TiO_2$@C electrode, respectively (see the determination of percentage volume in Supplementary Note 1). The ion-permeable membrane used in traditional RFBs to prevent cross-contamination[15,16,44–46] is not required here since the $Cl_2$-$CCl_4$ and $NaCl/H_2O$ are phase separated.

During charge, the $Cl_2$ was generated from oxidizing the $Cl^-$ in the $RuO_2$-$TiO_2$@C electrode. The reaction shows a constant potential at 1.2 V versus Ag/AgCl reference electrode [1.36 V versus normal hydrogen electrode (NHE)]. During discharge, the $Cl_2$ in $CCl_4$ was reduced to $Cl^-$ in the working electrode and entered the $NaCl/H_2O$ (see the formulation for positive electrode reaction). The presence of $CCl_4$ flow significantly enhances the coulombic efficiency (CE) from 8 to 97% (Fig. 1D). Because the solubility of $Cl_2$ in $CCl_4$ is three orders of magnitude higher than that in $NaCl/H_2O$ (0.184 mole/mole $CCl_4$ versus 0.0005 mole/mole $NaCl/H_2O$[38]) (Fig. 1E), the $Cl_2$ generated during the charging process can be stored in $CCl_4$, which prevents $Cl_2$ diffusion into $NaCl/H_2O$ as supported by Raman spectroscopy (Fig. S5) and the positive Gibbs free energy to transfer $Cl_2$ from $CCl_4$ to $NaCl/H_2O$ (Fig. S6). When 6.0 mL $CCl_4$ was used, a maximum reversible capacity for $Cl_2/Cl^-$ conversion is 600 mAh (Fig. S7), rendering the capacity of 97 Ah/L for the $Cl_2$-$CCl_4$.

*Positive electrode reaction.*

$$2Cl^- - 2e^- \leftrightarrow Cl_2 \qquad E^0 = 1.36\,V(\text{versus NHE})$$

The $Cl_2$-$CCl_4$ positive electrode has a low and almost consistent viscosity. When the concentration of $Cl_2$ increases from 0 to 0.184 mole/mole $CCl_4$ (saturation), the viscosity even slightly decreases from 0.894 to 0.819 mPa.s (Fig. 1F) in accord to Eyring's absolute reaction rate theory for gas–liquid mixtures[47,48]. On the other hand, the viscosity of common catholyte could increase by several or even dozen times as the concentration of solute increases[49]. The low viscosity of $Cl_2$-$CCl_4$ reduces the pumping loss[40], and the steady viscosity minimizes the

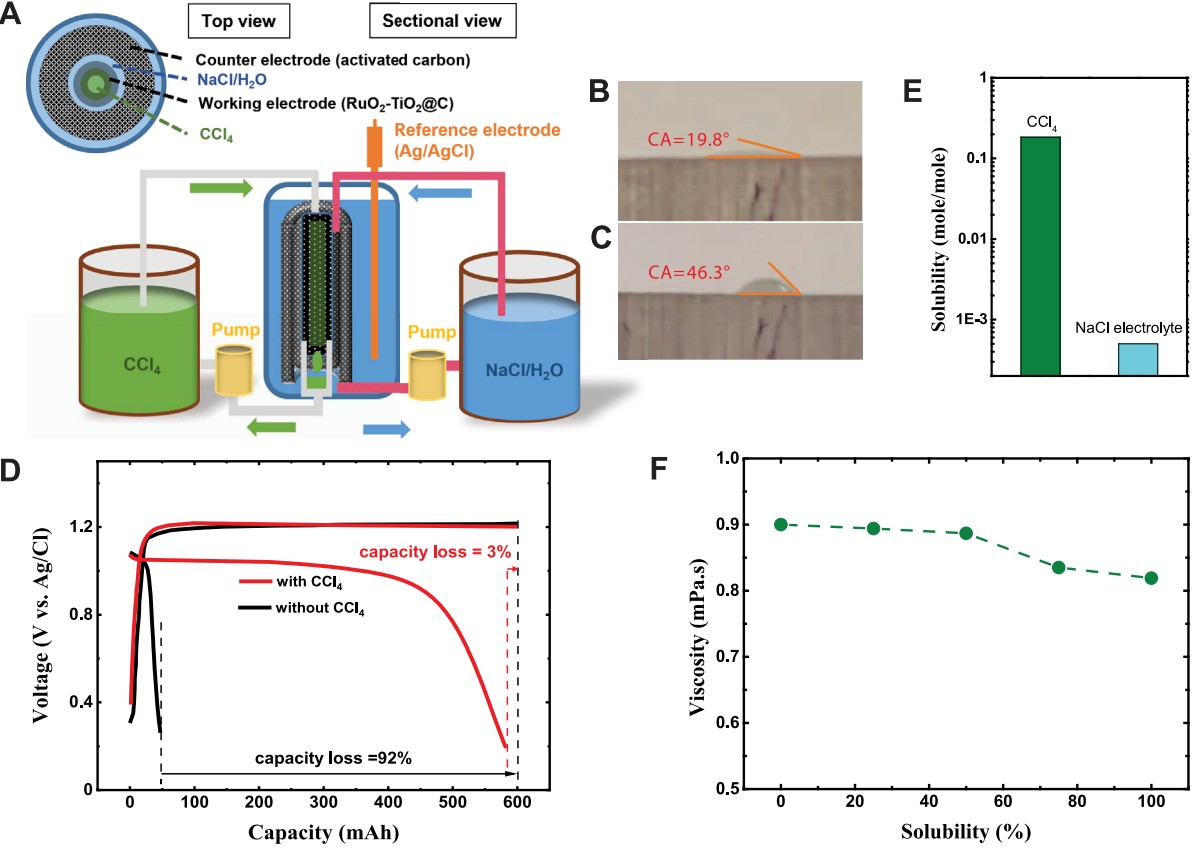

**Fig. 1 Electrochemical performance and physical properties of Cl₂-CCl₄. A** Schematic of the three-electrode cell. Inset shows the cylindrical structure of the cell from the top view, in which the inner diameter of the RuO₂-TiO₂@C working electrode is 2.0 mm, the thickness of the RuO₂-TiO₂@C electrode is 1.0 mm, the distance between the working and counter electrode is 3.0 mm and the thickness of the counter electrode is 3.0 mm. The height is 2.0 cm, and the volume capacity of the cell is around 2.0 mL. The total volumes of the CCl₄ reservoir and the NaCl/H₂O reservoir are 6.0 mL and 2.0 mL, respectively. **B** CA of CCl₄ on graphite plate electrode. **C** CA of NaCl/H₂O on graphite plate electrode. **D** Galvanostatic charge and discharge profiles of Cl₂-CCl₄ (red) and Cl₂ without CCl₄ (black) at the current density of 20 mA/cm². Both cells ran with constant charge capacity of 600 mAh at $Q_{aq}$ (flow rate of NaCl/H₂O) = 0.02 mL/s and $Q_{org}$ (flow rate of CCl₄) = 0.002 mL/s. The differences between discharge and charge capacity are labeled as percentage capacity loss. **E** The solubility of Cl₂ in CCl₄ and NaCl/H₂O. **F** The viscosities of Cl₂-CCl₄ with different concentrations of Cl₂ (100% refers to saturation).

volumetric transfer between catholyte and anolyte at different SOCs[50,51].

**Full chlorine flow battery (CFB).** To fabricate a full CFB, the activated carbon counter electrode was replaced by NaTi₂(PO₄)₃ negative electrode (Fig. 2A). NaTi₂(PO₄)₃ (Figs. S8, S9) was chosen as the negative electrode due to low potential (−0.5 V (versus NHE), rapid and reversible Na-ion insertion/extraction in NaCl/H₂O demonstrated by the symmetric anodic and cathodic peaks with 60 mV separation in the cyclic voltammetry (negative electrode reaction and Fig. S10A)[52]. The NaTi₂(PO₄)₃ shows a 65% capacity retention even at the C-rate of 315 C (1 C = fully discharge/charge within 1 hour, Fig. S10) and long cycle life of 1000 cycles (Fig. S11).

*Negative electrode reaction.*

$$Na_3Ti_2(PO_4)_3 - 2e^- - 2Na^+ \leftrightarrow NaTi_2(PO_4)_3 \quad E^0 = -0.5\,V(\text{versus NHE})$$

While the overpotentials enhanced (orange dash lines in Fig. 2B, C) as the current density increased, the discharge capacities did not vary (Fig. 2B, C), which could be attributed to the large reaction surface area endowed by the wetting between carbon electrode and Cl₂-CCl₄ (Fig. 1B, C). Fig. 2D demonstrates cell voltage efficiency (defined as the potential ratio of discharge

to charge) of 93.6% at 10 mA/cm² and ~77% at 100 mA/cm². The multiplication of discharge capacity and voltage gives the cell power density that peaks at 325 mW/cm² when operated at 350 mA/cm² (Fig. 2E). It is worth noting that polarizations for discharge are more significant than those for discharge (Fig. 2B, C). In the CFB, overpotentials are caused by redox reactions and concentration gradient. Since the symmetric factors for Cl⁻/Cl₂ redox reactions are equal[17,25], the overpotentials needed to drive the reduction and oxidation reaction are the same, the different overpotentials for charge and discharge observed here could only be attributed to the concentration gradient.

A steady-state model was developed to understand the species distribution and controlling steps in the CFB. The Nernst-Plank equation was applied to the porous RuO₂-TiO₂@C electrode (cell width = 0–1.0 mm in Fig. 2A), and NaCl/H₂O (cell width = 1.0–4.0 mm in Fig. 2A), Fick's equation was applied to the Cl₂-CCl₄ phase (cell width = −2.0–0 mm in Fig. 2A). The negative electrode was involved implicitly at the boundary of the NaCl/H₂O (cell width = 4.0 mm in Fig. 2A) (see model description and Tables S1–S4 in Supplementary Note 1). The model was validated by the agreement between the simulated and experimental cell voltages (black lines and dots in Fig. 3A, B, experimental potential retrieved from Fig. 2B, C) at the same flow rates and current densities.

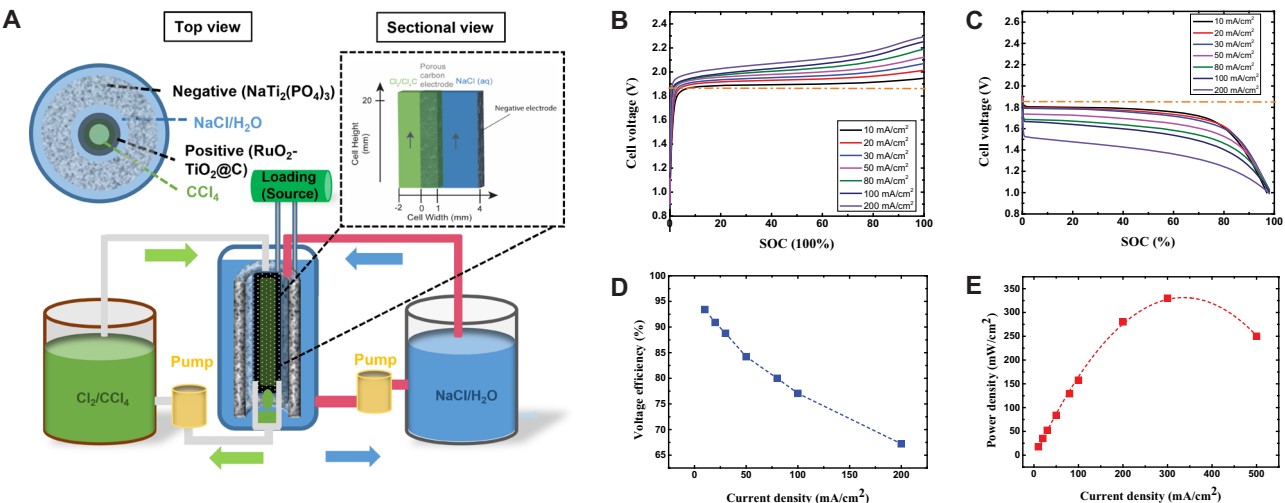

**Fig. 2 Schematic and electrochemical performance of chlorine flow battery (CFB). A** Schematic of the CFB, the inner diameter of the tube containing $CCl_4$ and $RuO_2$-$TiO_2$@C electrode is 2.0 mm, the thickness of the $RuO_2$-$TiO_2$@C electrode is 1.0 mm, the distance between the working and counter electrode is 3.0 mm. The thickness of the counter electrode is 3.0 mm. The height of the cell is 2.0 cm, and the volume capacity of the cell is around 2.0 mL. The total volumes of the $CCl_4$ reservoir and the $NaCl/H_2O$ reservoir are 6.0 mL and 2.0 mL, respectively. $Q_{aq}$ = 0.02 mL/s and $Q_{org}$ = 0.002 mL/s. Galvanostatic charge **B** and discharge **C** profiles of the CFB at different current densities. The state of charge (SOC) of the battery is normalized to the maximum reversible capacity at 10 mA/cm$^2$, in which 100% SOC represents charge to 600 mAh. **D** The voltage efficiencies of the CFB at different current densities. **E** The power densities of the CFB at different current densities.

The model was then used to visualize the species distribution in the $NaCl/H_2O$ and in $Cl_2$-$CCl_4$. During charge, the $Cl^-$ in $NaCl/H_2O$ was consumed inside the porous carbon electrode (Fig. 3C) and limits the reaction kinetics; during discharge, $Cl_2$ in $CCl_4$ is consumed and limits the reaction kinetics. The $Cl^-$ concentration gradients are more significant than the $Cl_2$ concentration gradient in the porous electrode for both charge and discharge (Fig. 3C, D), which is the result of a smaller diffusivity of $Cl^-$ ($1.5 \times 10^{-5}$ cm$^2$/s for $Cl^-$, $2.0 \times 10^{-5}$ cm$^2$/s for $Cl_2$ in $NaCl/H_2O$ and $3 \times 10^{-5}$ cm$^2$/s for $Cl_2$ in $CCl_4$[53–55] and lower volume percentage of $NaCl/H_2O$ than $CCl_4$ in the porous carbon electrode. The distinct species that control charge and discharge kinetics thus generate the asymmetric charge and discharge overpotentials (Fig. 2B, C). Since $Cl^-$ and $Cl_2$ are in different phases, increasing the flow rate of $NaCl/H_2O$ during charge and that of the $Cl_2$-$CCl_4$ during discharge enhance the mass transport of the limiting species accordingly, in which not only the overpotentials reduce, but the current density range allowing steady cell voltage extends (inset of Fig. 3A, B). At the highest flow rate examined, the voltage efficiency could be postulated to >93% at 20 mA/cm$^2$.

The high voltage efficiency of the cell is attributed not only to the fast reaction kinetics but also the membrane-free configuration. The potential gradient in the $NaCl/H_2O$ was determined by the model (Fig. 4A, B), and the potential difference across the cell at half-cell height was plotted. The potential drop of ~20 mV at 10 mA/cm$^2$ and ~250 mV at 100 mA/cm$^2$ (Fig. 4C) are equivalent to proton transport but over 5 times smaller than $Na^+$ and $K^+$ transport in Nafion ion-permeable membranes in aqueous flow batteries with similar cell dimensions[56]. Thus, removing the ion-selective membrane opens a range of chemistries to be investigated, as the charge carriers can be chosen arbitrarily.

The CFB demonstrates the round-trip energy efficiency of 91% (calculated by voltage efficiency × Coulombic efficiency) at 10 mA/cm$^2$ and provides an energy density of 125.7 Wh/L (see Methods), which is among the highest of the flow battery systems reported in past 10 years (Table S5). It is worth noting that the $Cl_2$-$CCl_4$ is different from bromine used in flow batteries that faces the serious self-discharge due to the diffusion of $Br_2$ to the negative electrodes in the form of polybromide. When ion-

permeable membranes were used to decrease $Br_2$ cross-over, voltage efficiency was significantly limited by the transport of ions in the membrane, resulting in <80% energy efficiency in overall performance[57–59]. Figure 5A, B show the measured cell voltage profile and stable round-trip cycling for this battery at 20 mA/cm$^2$ with a charge storage capacity of 600 mAh and the stable capacity retention for 500 cycles.

## Discussion

In this study, $CCl_4$ was used as a proof of concept, it can be replaced by other liquids with high $Cl_2$ solubility and are immiscible with $NaCl/H_2O$. The candidates include heptane (chlorine solubility = 0.173 mole fraction at ambient temperature), octane (chlorine solubility = 0.168 mole fraction at ambient temperature), tetradecane (chlorine solubility = 0.254 mole fraction at ambient temperature)[29] and mineral spirit. Mineral spirit demonstrates good wettability (CA = 9.1°) with carbon current collector (Fig. S12A), low viscosity (1.24 mPa.s), low toxicity, and is cheaper than $CCl_4$[60]. When $CCl_4$ was replaced by mineral spirit in the CFB, a volumetric capacity of 91.6 Ah/L was delivered at 20 °C (Fig. S12B).

The removal of the ion-permeable membrane also allows multivalent ions as charge carriers. When $ZnCl_2$ is added to the electrolyte, $NaTi_2(PO_4)_3$ can be replaced by zinc metal electrode, increasing the cell operating voltage to 1.9 V (Fig. S13).

Cost is one of the significant concerns to implementing flow batteries on a large scale for stationary energy storage. Considering that the ion-permeable membrane (mainly per-fluorinated polymers) takes up more than 30% of the cost of flow batteries, significant cost reduction is expected with the membrane-free design[20]. The total material cost for energy storage with the proposed CFB is estimated to be ~\$5/kWh, which is the cheapest among all the current flow battery systems (Fig. 5C and Table S5). In addition, the $RuO_2$ catalyst for chlorine evolution reaction (CER) can also be replaced by tin, zinc, cobalt, and other cheap metal oxides partially[30]. Therefore, the proposed CFB design leaves significant space to meet the stringent target of ~\$100/kWh for RFB applications[61].

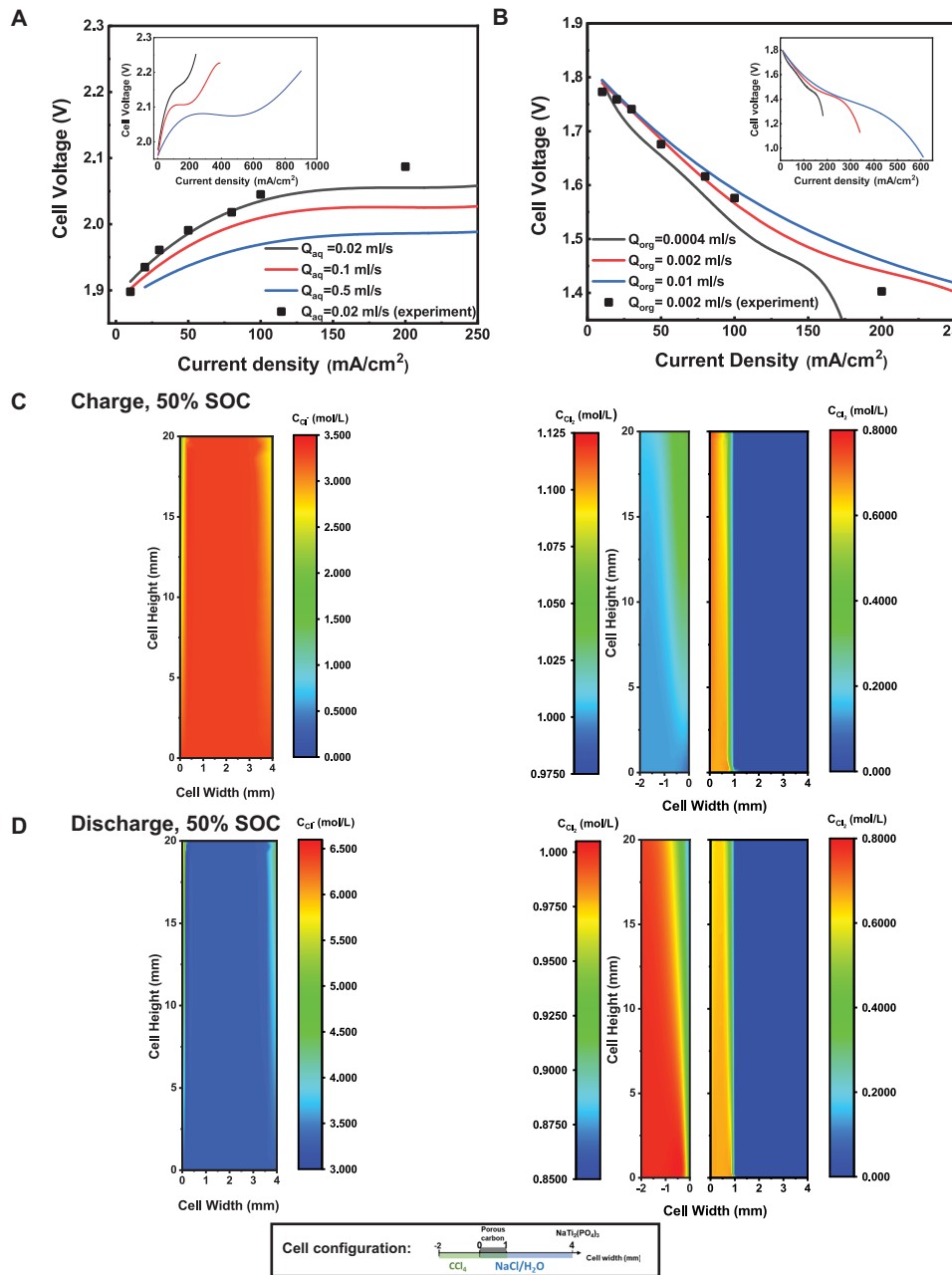

**Fig. 3 Simulation of the CFB. A** Steady-state potentials of CFB charged at 50% SOC with different $Q_{aq}$ and $Q_{org}$ = 0.002 mL/s, inset shows the whole current density range demonstrating steady charge potential. **B** Steady-state potentials of CFB discharged at 50% SOC with different $Q_{org}$ and $Q_{aq}$ = 0.02 mL/s, inset shows the whole current density range demonstrating steady discharge potential. **C** Distribution of $Cl^-$ and $Cl_2$ in the CFB charged at 50% SOC and 50 mA/cm$^2$ with $Q_{aq}$ = 0.02 mL/s and $Q_{org}$ = 0.002 mL/s. **D** Distribution of $Cl^-$ and $Cl_2$ in the CFB discharged at 50% SOC and 50 mA/cm$^2$ with $Q_{aq}$ = 0.02 mL/s and $Q_{org}$ = 0.002 mL/s. The position of $Cl_2$-$CCl_4$, $NaCl/H_2O$, porous $RuO_2$-$TiO_2$@C positive electrode and $NaTi_2(PO_4)_3$ negative electrode are labeled in the legend.

$Cl_2$ is a reactive chemical commodity used in paper, plastic, dye, textile, medicine, antiseptics, insecticide, solvent, and paint industries. Administration and engineering controls for storage and transport are available to confine the incident rate to 0.019% of total chlorine shipments between 2007 and 2017[62]. The Occupational Safety and Health Administration of the United States has set a permissible exposure limit at a time-weighted average of 0.1 ppm (0.68 mg/m$^3$) for bromine, 0.05 mg/m$^3$ for vanadium pentoxide dust, 0.1 ppm (0.4 mg/m$^3$) for quinone, and 1.0 ppm (3 mg/m$^3$) for chlorine[63]. Thus, there is no apparent increase in chemical exposure risk when changing to chlorine redox reaction.

However, protections and cautions are still crucial. The CFB proposed here is a closed system in which the leakage of $Cl_2$ gas is minimized by the fluoropolymer gasket (see Supplementary Note 2 for evaluation of chlorine permeation). Strategies from the chloro-alkali industry can be applied to reduce the risk of exposure upon scaling up, such as external seal pipe, shutoff system, neutralization reagents (scrubber)[64], and sensing systems[65].

In summary, the CFB proposed has demonstrated several unique advantages over current flow battery systems, including higher energy density, higher round-trip energy efficiency, and significantly lower prices. The membrane-free design enables both anionic and cationic charge carriers for a RFB, thus

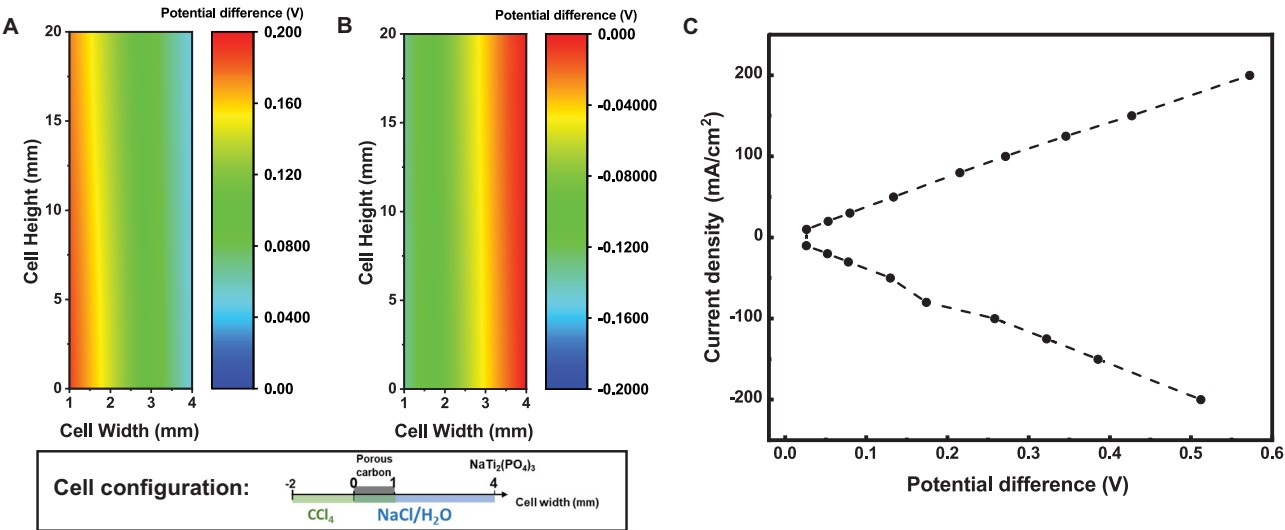

**Fig. 4 Potential gradient in the electrolyte of CFB. A** Potential distribution during charge and **B** during discharge at 50% SOC and 50 mA/cm². **C** The potential loss due to ion transport in the NaCl/H₂O at different current densities. In all cases $Q_{aq} = 0.02$ mL/s and $Q_{org} = 0.002$ mL/s. The positions of Cl₂-CCl₄, NaCl/H₂O, porous RuO₂-TiO₂@C positive electrode, and NaTi₂(PO₄)₃ negative electrode are labeled in the legend.

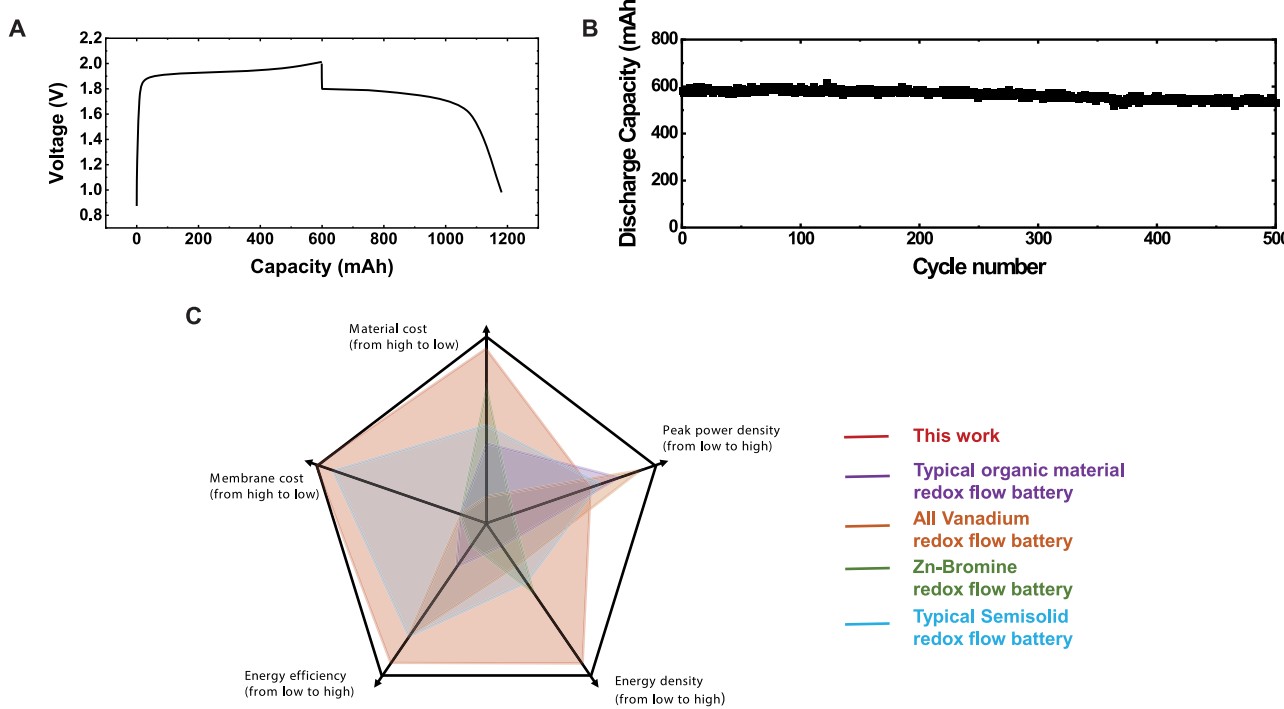

**Fig. 5 Charge and discharge behavior of CFB and comparisons of the performance matrices to redox flow batteries reported in the past 10 years. A** Cell voltage profiles during constant-current cycling and **B** cycle performance of CFB at 20 mA/cm², $Q_{aq} = 0.02$ mL/s and $Q_{org} = 0.002$ mL/s, and the charge capacity was set to be 600 mAh. The amount of CCl₄ is 6.0 mL, the size of the porous RuO₂-TiO₂@C electrode is 1.0 mm-thick and 2.0 cm² area. **C** The comparison of performance matrices among CFB, organic redox flow battery (anthraquinones as the anode material and ferricyanide as cathode material, ref. S24), all-vanadium redox flow battery (refs. S28, 29), Zn-Bromine redox flow battery (ref. S33), and semi-solid redox flow battery (Li as the anode and LiFePO₄ as cathode material ref. S34) (see details in Table S5).

expanding the material and chemistry space of the redox flow technologies.

## Methods

**Material synthesis**. The activated carbon with RuO₂/TiO₂ particles was prepared by dissolving 0.69 mmol RuCl₃ and 1.622 mmol C₁₆H₃₆O₄Ti in 100 mL isopropanol, then adding 2.0 g activated carbon into the solution. The mixture was stirred for 2 hours, and then the isopropanol was evaporated at 90 °C. Finally, the products were annealed at 500 °C for 1 hour under ambient conditions.

The carbon-coated NaTi₂(PO₄)₃ was synthesized from 0.002475 mol Na₂CO₃, 0.01485 mol NH₄H₂PO₄, and 0.0099 mol TiO₂ in 100 mL of a 2.0 wt% poly-vinyl-alcohol (PVA) aqueous solution. The mixture was stirred at 80 °C until the water evaporated and white solids formed. The white solids were placed in a porcelain boat and heated at 900 °C for 10 hours with an increasing rate of 5 °C/min under an N₂ flow in a tube furnace. To improve the cycling stability and electronic conductivity, thermal vapor deposition (TVD) was employed to prepare carbon-coated NaTi₂(PO₄)₃ after calcination. The as-prepared powder was transferred into a reaction tube to make a fluid-bed layer for the reaction at 700 °C for 2 hours where a toluene vapor was carried by N₂ through the reaction tube at a flow rate of

1 L/min, followed by heat-treatment at 900 °C for 2 hours without toluene carrying gas to increase its electronic conductivity. The temperature increasing rate is 5.0 °C/min.

**Electrode preparation and electrochemical measurements**. The working electrode was fabricated by pressing a mixture of the active materials (porous carbon or carbon-coated $NaTi_2(PO_4)_3$), carbon black, and PTFE (polytetrafluoroethylene) binder at the weight ratio of 7:2:1 onto a titanium grid with a pressure of 10 MPa. The cyclic voltammograms (CV) were obtained using a three-electrode cell with an active carbon counter electrode and Ag/AgCl reference electrode (0.197 V versus NHE). In the concentric cell, the inner diameter of the tube containing $CCl_4$ and the $RuO_2$-$TiO_2$@C electrode is 2.0 mm, the thickness of the porous carbon electrode is 1.0 mm, the distance between the counter and working electrodes is 3.0 mm, and the thickness of the counter electrode is 3.0 mm. The height of the cell is 2.0 cm, and the volume capacity of the cell is around 2.0 mL. The total volume of the $CCl_4$ reservoir is 6.0 mL, and the total volume of the $NaCl/H_2O$ reservoir is 2.0 mL. The CV measurements were carried out on a CHI660B electrochemical workstation. The galvanostatic charge and discharge profiles were obtained with an Arbin battery test station.

**Material characterizations**. Scanning electron microscopy (SEM) images were taken with Hitachi SU-70 analytical SEM (Japan). Powder X-ray diffraction (PXRD) data were collected on a Bruker D8 X-ray diffractometer using Cu Kα radiation ($\lambda = 1.5418$ Å). Raman spectroscopy was performed on a Horiba Jobin Yvon Labram Aramis using a 532 nm diode-pumped solid-state laser, attenuated to give ~900 μW power at the sample surface. Viscosity Measurements were carried out using a CANNON-FENSKE viscometer.

**Energy density calculations**. The energy density of CFB was calculated based on the 600 mAh cell used in this study with Eq. (1). The average operating potential is 1.8 V at 10 mA/cm², the volume of $CCl_4$ is 6.0 mL, the volume of $NaCl/H_2O$ is 2.0 mL and the volume of $Na(Ti_2(PO_4)_3)$ is 0.592 mL (weight = 5.0 g, density = 2.96 g/mL, volume = 2.96 g/mL ÷ 5.0 g = 0.592 mL). The total volume of active materials is 8.592 mL. Based on these configurations, the cell-level energy density (based on active materials) is 125.7 Wh/L.

$$\text{Energy density} = \frac{\text{Cell capacity} \times \text{average potential}}{\text{Total volume of active materials}} \quad (1)$$

## Data availability

The data that support the findings within this paper are available within the article and Supplementary Information. Additional data are available from the corresponding authors upon request. Source data are provided with this paper.

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

## Acknowledgements
This work was supported by the US Department of Energy ARPA-E Grant DEAR0000389.

## Author contributions
S.H., L.C., and Xiu.F. contributed equally to this work. S.H. and L.C. conceived the idea of a membrane-free chlorine flow battery. S.H. performed the numerical simulations and physiochemical measurements. L.C., Xiu.F., and Xiaotong.F. did the material synthesis and electrochemical measurements. X.J. performs the DFT calculations. S.H. and L.C. analyzed the results and wrote the manuscript. B.W., C.C., J.C., C.Y., W.W., and C.L. participated in the discussions. C.W. supervised all the studies.

## Competing interests
The authors declare no competing interests.
