## [Peer Review File · Nature Communications]

REVIEWER COMMENTS

Reviewer #1 (Remarks to the Author):

The paper is interesting because it brings the possibility of using an old system with re visited technology.

Abstract:

the abstract is information and easy to read with some useful performance data.

Would it be useful if the authors include keywords after the abstract?

Line 25; should it be strategy?

Lines 28-29; perhaps it would be fair to mention the other very important energy storage technologies.

Lines 30, 33, 36 and 38; it would be useful to include some performance data of the systems mentioned so the reader has an idea of the operational level in comparison with the one that it will be presented later.

Line 44; is the specific capacity only considers the redox species or includes electrolyte and cell components?

Line 45; state what is the values of the vanadium oxides RFB so the reader can compare and see the benefit of the proposed system.

Line 45; the energy density of all VRFB should be stated.

Line 46; how fast is the reaction in comparison to vanadium for example?

Line 51; is there any reference for the chloride/zinc battery from 1884?

Lines 60-69; what is the solubility of chlorine in CCl_4 ?

Line 66; what is the viscosity of the aqueous vanadium for example just for comparison?

Line 64; the specific capacity given for the $\text{Cl}_2\text{-CCl}_4$ is given in litres, should not be called volumetric capacity? Also, how it compares to the vague given in line 44?

In general, what would be the volatility of chlorine from the tetrachloride to the environment? Would there be any danger of dispersing chlorine gas into the atmosphere for example if the cell increases its temperature?

What are the risk of chlorine gas leaking to the environment?

Figures 1A and S2 should include the dimensions.

Lines 75-76; how robust would be carbon substrate for long term operations?

The arrows indicating the direction of the flow in figure 1A should be larger to clarify where in the electrode the electrolyte is been pumped.

Figures 1B and 1C are not explained; can you clarify the objective of the drop seen in 1C, what is the surface, the wettability? Include labels to inform of the materials.

Line 84; the charge discharge reactions should be included?

Line 92; explain why the CE drops to 8% with no CCl₄.

Figure 1A; what does the blue colour layer next to the electrodes represents? Is the NaCl/H₂O electrolyte? Why it is only on the surface of the electrodes? What is the whit/black grid in between the electrodes?

Figure 1D; what is the difference between the two red curves and the difference between the two black curves?

Line 92; since there is some solubility of Cl₂ in the NaCl electrolyte, even if it is very low compared to the solubility in CCl₄, is there any chloride reaction detected in the counter electrode?

what are the reactions in the counter electrode during charge / discharge cycles?

Line 99; is saturation 0.184?

Line 100; why not state the system that shows an increase of viscosity and the positive or negative impact of this property on the cell performance?

Line 105; should not be contact angle? Please revise through the manuscript.

Lines 104-111; it would be beneficial to state the dimensions of the cell and the volume capacity of the cell and the reservoirs. If it is in the supplementary section, state it.

Full Chlorine battery section:

Line 118; 315 C is charge, not current density. Why is it referred as current density, please revise.

Line 126; the authors refer to Figures 2B and 2C to indicate constant CE Current Efficiency?) of 97% however Figure 2B and 2C show cell and the cell voltage versus SOC; please clarify what do they want to say. Also, the label B appears as if it refers to the enhancement part of the cell; the B label should be on the graph.

Lines 127-131; compare the values of this battery with similar systems from the literature and highlight the advantages of the CFB system.

Line 143; if the difference in the overpotentials for charge and discharge could only be attributed to mass transport effect, what is the cause?

Line 160; please explain why the difference in chloride ion concentrations is larger than chlorine gas in the porous electrodes.

The axes of the inset curves in figures 3A and 3B should have labels or specify that they are the same as the main figure if they are.

Line 181; is there any way to provide data about how fast the kinetic rates are, even from the literature?

Figures 3C, 3D 4A and 4B should include labels to indicate the position of the anode and cathode.

Line 197; what is it included in the volume to calculate such value of the volumetric energy density?

Line 200. Include the word "the" before anode. Please revise through the document for similar missed articles, i.e. "the".

Discussion section

Line 218-219; mention the other liquids that can replace CCl₄.

Line 223; correct the units after 20

Line 235; correct the error in the references after the words "free design".

Some references (36, 37) only mention the first author, et al. Is there any guideline for the numbers of authors or why not all the authors are named? Would not be fair to include all the authors that contributed to a certain reference?

References to major revisions of RFB are missing for example:

<https://DOI.org/10.1021/bk-2020-1364.ch001>

<https://DOI.org/10.1039/c2ra21342g>

<https://doi.org/10.1016/j.jpowsour.2006.02.095>

Reviewer #2 (Remarks to the Author):

Chen et al report a non-aqueous flow battery that does not require a membrane/separator due to the fact it has two immiscible (water and carbon tetrachloride) solvents. The battery also has a second significant advancement with facile molecular chlorine and chloride electrochemistry. This is a very elegant and thorough study. I am also quite impressed by the thoroughness of the supporting information. I have worked on non-aqueous flow battery research and development for more than ten years and I consider this to be in the top 10% most significant advancements. The major Achilles heel is my concern about the safety of using chlorine gas in a grid-scale (large-scale) battery. Safety is briefly mentioned right at the end of the manuscript, but I think it is a big mistake to just gloss over it, particularly with the broad readership of Nature Communications. A minor point is that the

sentence structure (especially at the beginning of the manuscript) is awkward in places and could use some improvement.

Reviewer #3 (Remarks to the Author):

Comments : In this paper, a new cell structure of membrane-free chlorine flow battery with high energy density and low-cost was developed. Both high solubility of Cl₂ in CCl₄ and immiscibility between CCl₄ and NaCl electrolyte enable (i) excellent flow battery cell performance with extremely high energy density, and (ii) membrane-free design durable even at high current density. Additional mechanical modification that was carried out by adjusting flow rate could play a role in manipulating the unbalance of flow battery occurring by the difference in flowing speed accelerated by different phase of CCl₄ and NaCl. However, there are some points to be more clarified.

[1] In page 3, there is the sentence that “As the concentration of Cl₂ increases from zero to saturation, the viscosity even slightly decreases from 0.894 mPa.s to 0.819 mPa.s, while the viscosity of common catholyte could increase for several or even dozen times as the concentration of solute increases.”. Authors should explain this result more specifically because they only mentioned the result itself. Especially, there is no explanation about how the viscosity of electrolytes became lower with increased concentration of solute such as Cl₂.

[2] In page 5, there is the sentence that “Since Cl⁻ and Cl₂ are in different phases, the reaction kinetics of the cell can be further improved by increasing the flow rate of NaCl electrolyte during charge and that of the CCl₄ during discharge, ~.” The tested flow rate of NaCl electrolyte was 0.02, 0.1, 0.05 mL/s, and that of CCl₄ electrolyte was 0.0004, 0.002, 0.001 mL/s. Authors should explain why authors chose the range of flow rate. In addition, authors should explain the effect of optimal flow rate on the performances of flow battery cell because its cell voltage is sensitively affected by the flow rate.

[3] In page 8, Fig. 5 shows the cell performance of flow battery cell during 50 cycles. However, this is too short because the NaTi₂(PO₄)₃ electrode takes advantage of long cycle life (1,000 cycles) as authors mentioned in this manuscript, Therefore, authors should include more long-cycling cell performance (>500 cycles) data to ensure the long life time of this flow battery cell system. As authors know, the long life time is important factor for flow battery cell system .

[4] In the introduction part, I recommend you to add the paragraph explaining organic redox flow batteries as the example of one of promising RFB types with the following related references.

- (2020). Substituent pattern effects on the redox potentials of quinone-based active materials for aqueous redox flow batteries. *ChemSusChem*, 13(20), 5480-5488.

- (2020). Alkaline aqueous organic redox flow batteries of high energy and power densities using mixed naphthoquinone derivatives. *Chemical Engineering Journal*, 386, 123985.

- (2019). Extending the lifetime of organic flow batteries via redox state management. *Journal of the American Chemical Society*, 141(20), 8014-8019.
- (2020). Tailoring Dihydroxyphthalazines to Enable Their Stable and Efficient Use in the Catholyte of Aqueous Redox Flow Batteries. *Chemistry of Materials*, 32(8), 3427-3438.
- (2018). Alkaline quinone flow battery with long lifetime at pH 12. *Joule*, 2(9), 1894-1906.

REVIEWER COMMENTS on “High Energy and Low-cost Membrane-free Flow Battery”

Reviewer #1 (Remarks to the Author):

The paper is interesting because it brings the possibility of using an old system with re-visited technology.

We want to express the gratitude on the reviewers' time and effort to help improve this manuscript. The listed issues are addressed point-by-point.

Abstract:

the abstract is information and easy to read with some useful performance data.

1. Would it be useful if the authors include keywords after the abstract?

Reply: The keywords were added after the abstract in the revised main text and highlighted as follow:

“Keyword: Redox flow batteries, chloride-chlorine conversion, membrane-free”.

2. Line 25; should it be strategy?

Reply: The sentence was corrected and highlighted in the main text as follow:

“The integration of renewable energy, such as the solar and wind power, is the most important strategy to reduce the carbon emission for sustainable development.”

3. Lines 28-29; perhaps it would be fair to mention the other very important energy storage technologies.

Reply: Other grid scale energy storage technologies were mentioned and highlighted in the revised main text as follow:

“Different technologies, such as supercapacitors, sodium-sulfur batteries, pump hydro, flywheels, superconducting magnetic energy and so forth are developed for this purpose⁵, among which redox flow battery (RFB) is considered one of the most attractive for large-scale energy storage due to the lower capital cost, higher energy conversion efficiency and facile modularity^{6,7}.”

4. Lines 30, 33, 36 and 38; it would be useful to include some performance data of the systems mentioned so the reader has an idea of the operational level in comparison with the one that it will be presented later.

*Reply: We included a detailed comparison of RFBs based on different redox chemistries (inorganic, organic, polymer and slurry active materials) that published in the recent 10 years in **Table S5**. The comparisons involve performance data such as energy density, power density and energy efficiency. To avoid repetition, we will not elaborate here.*

5. Line 44; is the specific capacity only considers the redox species or includes electrolyte and cell components?

Reply: The specific capacity discussed here is only based on the weight of Cl₂ and vanadium oxides only. The purpose is to compare the theoretical capacity provided by the active material without complication by cell configuration and electrolyte design.

6. Line 45; state what is the values of the vanadium oxides RFB so the reader can compare and see the benefit of the proposed system.

Reply: The specific capacity of vanadium oxides redox couple VO₂⁺/VO²⁺ used in RFBs is 226 mAh/g considering the following redox reaction (the weight of protons is included).

This information is supplied and highlighted in the revised main text as follow:

“...we report a new RFB system capitalizing on the electrolysis of aqueous NaCl electrolyte, that features the Cl₂/Cl⁻ redox couple with a theoretic capacity of 755 mAh/g as active material for positive electrode, which is more than 2 times that of vanadium oxides (VO₂⁺/VO²⁺, 226 mAh/g) used in current RFBs.”

7. Line 45; the energy density of all VRFB should be stated.

Reply: The energy density is multiplication of cell voltage and capacity. The cell voltage of all VRFB is the difference between catholyte and anolyte potentials (**Table R1**), which is 1.246 V. The capacity is the electron transferred per liter catholyte and anolyte. When 1.0 liter 1.0 M VO₂⁺ and 1.0 liter 1.0 M V²⁺ anolyte are used, the cell capacity= 96500 C/ 2 liter = 13.4 Ah/L. When 1.0 liter saturated VO₂⁺ (1.8 M VO₂⁺ in 3.0 M H₂SO₄ supporting electrolyte at 20 °C (1)) is used (assuming the concentration of V²⁺ anolyte is the same), the cell capacity raises to 24.1 Ah/L. And the energy density of all VRFBs at these two concentrations are 16.7 Wh/L and 30.0 Wh/L respectively.

While this information is not provided in the main text, the energy density of VRFB using HCl supporting electrolyte with an overall higher energy density was listed in **Table S5** (*Adv. Energy Mater.* **1**, 394–400 (2011)), thus we will not repeat here.

Table R1. The formula and potentials of the redox reactions in anolyte and catholyte of all vanadium redox flow batteries.

Anolyte:	$V^{3+} + e^- \leftrightarrow V^{2+}$	$E^0 = -0.255 \text{ V (versus NHE)}$
Catholyte:	$VO_2^+ + 2H^+ + e^- \leftrightarrow VO^{2+} + H_2O$	$E^0 = +0.991 \text{ V (versus NHE)}$

8. Line 46; how fast is the reaction in comparison to vanadium for example?

Reply: The reaction kinetics for Cl⁻/Cl₂, V²⁺/V³⁺ and VO₂⁺/VO²⁺ can be characterized by reaction activation energy (E_a). In particular, the exchange current densities of these reactions at different temperatures were measured and the E_a can be obtained by fitting the exchange current densities with Arrhenius equation. E_a for Cl⁻/Cl₂ redox on RuO₂-TiO₂ in 4.0 M NaCl water solution and at atmospheric pressure is 35.5 kJ/mol (8.5 kcal/mol) (2), E_a for VO₂⁺/VO²⁺ in 0.2 M VO²⁺ with 0.5 M H₂SO₄ supporting electrolyte varies from 45 to 70 kJ/mol from 0% to 100 % SOC (3). The results are added and highlighted in the revised main text as follow:

“ Cl₂/Cl⁻ redox chemistry is a fast single-electron transferred reaction with an activation energy of 35.5 kJ/mol^{23,24}, which is comparable or even smaller than that of VO₂⁺/VO²⁺ ²⁵, thus is suitable for high power applications. ”

9. Line 51; is there any reference for the chloride/zinc battery from 1884?

Reply: The reference was added and highlighted in the revised main text as reference 28:

“28. Winter, Lumen & Degner, Glenn, Minute Epics of Flight, New York, Grosset & Dunlap, 1933.”

And it is called out in the revised main text as following:

“Rarely heard in the battery history, Cl₂/Cl⁻ redox couple with the Zinc negative electrode yet was in fact invented as the first redox flow battery to power the first fully controlled airship La France in 1884²⁸.”

10. Lines 60-69; what is the solubility of chlorine in CCl₄?

Reply: The solubility of chlorine in CCl₄ is 0.184 mole/ mole CCl₄ at 20 °C (4) . It was highlighted in the revised main text as following:

“Cl₂-CCl₄ delivers a volumetric capacity of 97 Ah/L due to high solubility of Cl₂ in CCl₄ (0.184 mole/mole CCl₄³⁸), which is 2 to 4 times improvement over the current vanadium-based catholyte (22.6 - 43.1 Ah/L³⁹)”

11. Line 66; what is the viscosity of the aqueous vanadium for example just for comparison?

*Reply: The viscosities of electrolytes used in all VRFBs vary with the concentration of vanadium oxides, the supporting electrolytes and temperature. Since it is infeasible to measure the viscosities for all possible electrolyte compositions, approximations based on the VOSO₄ solution and specified supporting electrolyte are generated using viscosity models. Eyring's model was demonstrated to provide accurate prediction for VOSO₄ in H₂SO₄ electrolyte. Here we provide the viscosity of VOSO₄ aqueous solution at different concentrations in **Table R2 (results regenerated from (5))** and the predicted viscosity of VOSO₄ in 1.5 M H₂SO₄ electrolyte in **Table R3 (results regenerated from (5))**. It can be clearly seen that the viscosity of VOSO₄ solution raises by 3 to 4 times at room temperature when the concentration changes from 0.5 m to 2.2 m (**Table R2**), which corresponds to the concentration between 25% and 100% SOC in VRFBs. With H₂SO₄ supporting electrolyte, the variation of viscosity is less obvious, but the overall values increase (**Table R3**). Based on this information, the main text was modified and highlighted as following:*

“ The Cl₂-CCl₄ has low and constant viscosity of 0.819 mPa.s, in contrast to higher and varying viscosities of vanadium oxide electrolytes (1.4 -3.2 mPa.s⁴⁰), thus is easy to flow. ”

Table R2. The viscosity of $VOSO_4$ aqueous solution at different concentrations and temperatures. The concentrations labelled in red are higher than those commonly used in VRFBs (results regenerated from (5)).

C_{VOSO_4} (m/mol kg ⁻¹)	Viscosity (mPa·s)								
	283.15 K	288.15 K	293.15 K	298.15 K	303.15 K	308.15 K	313.15 K	318.15 K	323.15 K
0.5	1.8681	1.6180	1.4158	1.2508	1.1129	0.9984	0.9018	0.8193	0.7506
0.6986	2.1512	1.8571	1.6197	1.4253	1.2649	1.1312	1.0186	0.9230	0.8422
0.8956	2.5269	2.1682	1.8831	1.6488	1.4575	1.2993	1.1657	1.0523	0.9558
1.0009	2.7760	2.3786	2.0601	1.8032	1.5899	1.4102	1.2583	1.1282	1.0211
1.1991	3.1856	2.7205	2.3484	2.0473	1.8003	1.5959	1.4241	1.2798	1.1566
1.3927	3.7343	3.1637	2.7197	2.3616	2.0694	1.8279	1.6269	1.4567	1.3131
1.5806	4.3344	3.6614	3.1345	2.7102	2.3653	2.0819	1.8461	1.6476	1.4799
1.8025	5.2489	4.4063	3.7447	3.2179	2.7940	2.4487	2.1598	1.9205	1.7186
1.9964	6.1267	5.1179	4.3333	3.7091	3.2035	2.7956	2.4604	2.1799	1.9444
2.1951	7.2237	5.9960	5.0470	4.2961	3.6978	3.2149	2.8176	2.4890	2.2129
2.3667	8.3527	6.9012	5.7804	4.9010	4.1983	3.6344	3.1767	2.7984	2.4778
2.5913	10.092	8.2703	6.8832	5.8007	4.9461	4.2585	3.7055	3.2534	2.8759
2.7973	12.049	9.8082	8.1042	6.7936	5.7638	4.9438	4.2785	3.7381	3.2907
2.9953	14.114	11.787	9.6735	8.0654	6.8020	5.8042	5.0035	4.3567	3.8186

Table R3. The viscosity of $VOSO_4$ in 1.5 M H_2SO_4 supporting electrolyte at different concentrations and at 298.15K predicted by Eyring's absolute reaction rate theory (results regenerated from (5)).

C_{VOSO_4} (m/mol kg ⁻¹)	Viscosity (mPa·s)		
	283.15 K	298.15 K	308.15 K
0.2985	2.0732	1.4003	1.1250
0.7971	2.8387	1.8639	1.4743
1.6495	5.1349	3.1766	2.4403

12. Line 64; the specific capacity given for the Cl₂-CCl₄ is given in litres, should not be called volumetric capacity? Also, how it compares to the vague given in line 44?

Reply: The sentence has been corrected and highlighted in the main text as following. To obtain the volumetric capacity of chlorine gas, the specific capacity in line 44 is multiplied by density of chlorine gas at room temperature = 775 mAh/g 3.04 g/L =2.295 Ah/L. The lower volumetric capacity of chlorine gas versus Cl₂-CCl₄ is due to the low density of gas.*

“Cl₂-CCl₄ delivers a volumetric capacity of 97 Ah/L...”

13. In general, what would be the volatility of chlorine from the tetrachloride to the environment? Would there be any danger of dispersing chlorine gas into the atmosphere for example if the cell increases its temperature?

Reply: The cell is a closed system with the Cl₂-CCl₄ contained in glass container assembled under atmospheric pressure, the leakage could potentially occur at the seal through either chemical corrosion or permeation. The sealing gasket for the bottle and the tubing for Cl₂-CCl₄ flow consists of Viton fluoroelastomer, which is resistive to corrosion from chlorine and swelling by halogenated solvents demonstrating less than 10% volume expansion (6) in a wide temperature range between -20 °C to 200 °C, thus we only consider the leakage through Cl₂ permeation.

*Cl₂ permeation rate was calculated with **equation R1** (7), which depends on the difference of Cl₂ partial pressure (p_{Cl₂}) inside and outside the gasket. The partial pressure of Cl₂ for Cl₂-CCl₄ at different concentrations can be calculated with Raoult's law for binary mixture. The p_{Cl₂} inside the container is 111.22 kPa when concentration of Cl₂ is 0.184 mol/mol CCl₄ (saturation) at 20 °C . At 50 °C, the p_{Cl₂} of saturated Cl₂-CCl₄ increases to 116.0 kPa (0.088 mol Cl₂/mol CCl₄) (**Table R4**). The p_{Cl₂} outside the container is assumed to be 0 kPa, the permeation flux of Cl₂ through the Viton gasket (F) into the atmosphere calculated through **equation R1** is between 1.24- 19.03 mL/day (3.72-57.1 mg /day) at 20 °C and 2.08- 30.60 mL/day (5.1-75 mg/day) at 50 °C. (The variations are caused by the different permeation coefficients reported for different fluoroelastomers (8)). At these leakage rates, it requires 0.5 to 20 days at 20 °C or 0.3 to 15 days at 50 °C to reach the permissible exposure limit (PEL) of chlorine (3mg/ m³ (9)) in an unventilated 1 m² * 2 m storage space (around the size of a fume hood). Thus, with the appropriate gaskets, chlorine sensor and in a ventilated environment, it is very unlikely to be exposed to chlorine higher than the PEL.*

$$F = K \frac{A(p_{Cl_2,in} - p_{Cl_2,out})}{d} \quad (R1)$$

*F= permeation flux of the gas (mL/day) ; K= permeation coefficient=60- 1000 (20 °C), 100 -2000 (50 °C) (mL/100 in²/day/atm) (8) ; A= gasket surface area= (1.5 cm*1.5 cm-0.5 cm* 0.5 cm)*π*2 =12.56 cm²; d= gasket thickness= 1 cm; p_{Cl₂,in} = partial pressure of Cl₂ inside the container; p_{Cl₂,out}= partial pressure of Cl₂ outside the container*

Table R4. Vapor pressures of Cl_2 , CCl_4 and $\text{Cl}_2\text{-CCl}_4$ at $20\text{ }^\circ\text{C}$ (atm) and $50\text{ }^\circ\text{C}$ (atm).

Pressure (kPa)	P_{Cl_2} (kPa)	P_{CCl_4} (kPa)	p_{Cl_2} (kPa) (calculated with Raoult's law)
$20\text{ }^\circ\text{C}$ (1 atm)	604.50 (10)	13.33 (11)	$p_{\text{Cl}_2} = P_{\text{Cl}_2} \times x_{\text{Cl}_2} = 111.22$ ($x_{\text{Cl}_2} \leq 0.184$)
$50\text{ }^\circ\text{C}$ (1 atm)	1318.97 (10)	53.33 (11)	$p_{\text{Cl}_2} = P_{\text{Cl}_2} \times x_{\text{Cl}_2} = 116.0$ ($x_{\text{Cl}_2} \leq 0.088$)

The discussion above is supplied in the Supplementary Notes section in the revised supplementary materials.

14. What are the risks of chlorine gas leaking to the environment?

Reply: As discussed in question 13, the permeability of Cl_2 across Viton gasket at $20\text{ }^\circ\text{C}$ and $50\text{ }^\circ\text{C}$ is not significant.

15. Figures 1A and S2 should include the dimensions.

Reply: The dimensions of the cell and set up were added to the captions of **Fig. 1A** and **S1** (S2 in original manuscript).

16. Lines 75-76; how robust would-be carbon substrate for long term operations?

Reply: As shown in **Fig. R1** that the morphology of the activated carbon maintains after 500 cycles, indicating that robustness of the carbon substrate in this system.

Fig R1. SEM images of activated carbon after 500 cycles.

17. The arrows indicating the direction of the flow in figure 1A should be larger to clarify where in the electrode the electrolyte has been pumped.

Reply: The arrows in Fig. 1A have been enlarged to demonstrate the flow directions as shown in Fig. R2. Fig. 1A was replaced by Fig. R2 in the revised main text.

Fig. R2. Schematics of the three-electrode cell, inset shows the cylindrical structure of the cell from the top.

18. Figures 1B and 1C are not explained; can you clarify the objective of the drop seen in 1C, what is the surface, the wettability? Include labels to inform of the materials.

Reply: Fig. 1B and 1C show the contact angle measurement. 5 μ L of CCl₄ and NaCl aqueous solution were dropped onto the graphite plate electrode in Fig. 1B and 1C correspondingly to determine the wettability of these liquids to carbon. The contact angles (CA) are less than 90 degrees for both liquids with CCl₄ demonstrates smaller CA, suggesting that they both wet carbon substrate but the CCl₄ has higher wettability. This information is supplied and highlighted in the revised main text as following:

“ While CCl₄ and NaCl electrolyte entered RuO₂-TiO₂@C electrode as separate flows, they both wet the carbon surface as demonstrated by less than 90° contact angles for both liquids on a graphite plate electrode (Fig. 1B and Fig. 1C) and takes up 66.2% and 33.8% of the void volume in the RuO₂-TiO₂@C electrode, respectively (see determination of percentage volume in Supplementary Materials). ”

19. Line 84; the charge discharge reactions should be included?

Reply: The charge and discharge reactions of the Cl₂/Cl⁻ positive electrode and the corresponding potential are listed below. This formula is listed and highlighted in the revised main text.

Positive electrode reaction: $2\text{Cl}^- - 2e^- \leftrightarrow \text{Cl}_2$

$E^0 = 1.36 \text{ V (versus NHE)}$

20. Line 92; explain why the CE drops to 8% with no CCl₄.

Reply: The reason that the CE drops to 8% with no CCl₄ is because the Cl₂ gas generated during charge process cannot be stored in the electrolyte, then the escaped Cl₂ cannot be converted back to Cl⁻ during the discharge process, resulting in low CE. The discussion is added and highlighted in the revised main text as following:

“The presence of CCl₄ is essential for the reversibility, demonstrated by the significant enhancement of the Coulombic efficiency (CE) from 8% without CCl₄ flow to 97% with CCl₄ flow (Fig. 1D). The reason is that the Cl₂ can be stored in CCl₄ much more efficiently than in NaCl/H₂O electrolyte due to a three orders of magnitude difference in solubility (0.184 mole/mole CCl₄ versus 0.0005 mole/mole NaCl/H₂O electrolyte³⁸)(Fig. 1E). Without CCl₄, the Cl₂ generated during charge process leaves the electrolyte and cannot be converted back to Cl⁻ during discharge.”

21. Figure 1A; what does the blue color layer next to the electrodes represents? Is the NaCl/H₂O electrolyte? Why it is only on the surface of the electrodes? What is the whit/black grid in between the electrodes?

Reply: The blue color layer is the NaCl/H₂O electrolyte. The white/black grid is the counter electrode, and it is submerged in the NaCl/H₂O electrolyte (blue layer). In order to avoid confusion, inset of Fig. 1A was modified as in Fig. R2 to better show the labels of each component in the cell.

22. Figure 1D; what is the difference between the two red curves and the difference between the two black curves?

Reply: The capacity difference between the two red curves is 18 mAh, corresponding to 3% capacity loss in a 600 mAh cell, and the capacity difference between the two black curves is 553 mAh, corresponding to 92% capacity loss (Fig. R3). Fig. 1D was replaced by Fig. R3 and highlighted in the revised main text.

Fig. R3. Galvanostatic charge and discharge profiles of $\text{Cl}_2\text{-CCl}_4$ (red) and Cl_2 without CCl_4 (black) at the current density of 20 mA/cm^2 . Both cells ran with constant charge capacity of 600 mAh at Q_{aq} (flow rate of NaCl electrolyte) = 0.02 mL/s and Q_{org} (flow rate of CCl_4) = 0.002 mL/s. The difference between discharge and charge capacity are labeled as percentage capacity loss.

23. Line 92; since there is some solubility of Cl_2 in the NaCl electrolyte, even if it is very low compared to the solubility in CCl_4 , is there any chloride reaction detected in the counter electrode? what are the reactions in the counter electrode during charge/ discharge cycles?

Reply: The counter electrode in the three-electrode cell is activated carbon electrode that undergoes adsorption and desorption of Na^+ during charging and discharging the $\text{Cl}_2\text{-CCl}_4$ cathode. No chloride reaction was detected, due to negligible Cl_2 cross-over as demonstrated in Fig. S5.

24. Line 99; is saturation 0.184?

Reply: Yes, the Cl_2 concentration is 0.184 mole/mole CCl_4 in saturated solution (4). To clarify, the sentence is changed and highlighted in the revised main text as following:

“When the concentration of Cl_2 increases from 0 to 0.184 mole/mole CCl_4 (saturated concentration), the viscosity even slightly decreases from 0.894 mPa.s to 0.819 mPa.s (Fig. 1F), which is in accord to the Eyring’s absolute reaction rate theory for gas-liquid mixture^{48,49}.”

25. Line 100; why not state the system that shows an increase of viscosity and the positive or negative impact of this property on the cell performance?

Reply: The increase of viscosity has several drawbacks on the battery performance. In the cases of solid slurry or polymer based RFBs, the higher viscosity of catholyte/anolyte will cause larger pressure-drop along the flow direction in the cell or reduce flow rate at the same pressure drop, thus the concentration polarization cannot be easily alleviated as in traditional RFBs and the

peak power is limited (12). To achieve the same flow rate, the energy and cost consumption on pumping must be increased.

In all-vanadium or vanadium-iron hybrid redox flow batteries, the overall viscosities of the electrolytes are much smaller, but the viscosities of catholyte and anolyte change differently when the SOC changes, causing unbalanced osmotic pressure across the ion-exchange membrane (13) and volumetric transfer (or cross-over) between catholyte and anolyte (14). Thus, the low and steady viscosity of CCl_4 prevent these undesired situations from happening. The discussion was added and highlight in the revised main text as following:

“The low viscosity of $\text{Cl}_2\text{-CCl}_4$ reduces the pumping loss⁴⁰ and the steady viscosity minimizes the unbalanced osmotic pressure and volumetric transfer between catholyte and anolyte at different SOCs^{51,52}.”

26. Line 105; should not be contact angle? Please revise through the manuscript.

Reply: Thank you for the comment. The typos have been revised through the manuscript.

27. Lines 104-111; it would be beneficial to state the dimensions of the cell and the volume capacity of the cell and the reservoirs. If it is in the supplementary section, state it.

*Reply: The inner diameter of tube containing CCl_4 and $\text{RuO}_2\text{-TiO}_2\text{@C}$ working electrode is 2 mm, the thickness of the $\text{RuO}_2\text{-TiO}_2\text{@C}$ electrode is 1 mm, the distance between the working and counter electrode is 3 mm and the thickness of the counter electrode is 3 mm. The height of the cell is 2 cm and the volume capacity of cell is around 2 mL. The total volumes of the CCl_4 reservoir and the $\text{NaCl/H}_2\text{O}$ reservoir are 6 mL and 2 mL, respectively. These configurations were added and highlighted in the caption of **Fig. 1A** in the revised main text and Methods section in the Supplementary Materials.*

Full Chlorine battery section:

28. Line 118; 315 C is charge, not current density. Why is it referred as current density, please revise.

Reply: The sentence has been changed and highlighted in the revised main text as following:

*“The $\text{NaTi}_2(\text{PO}_4)_3$ shows a 65% capacity retention even at a high C-rate of 315 C (1C = fully discharge/charge with 1 hour, **Fig. S10**), and long cycle life of 1000 cycles (**Fig. S11**)”*

29. Line 126; the authors refer to Figures 2B and 2C to indicate constant CE Current Efficiency?) of 97% however Figure 2B and 2C show cell and the cell voltage versus SOC; please clarify what do they want to say. Also, the label B appears as if it refers to the enhancement part of the cell; the B label should be on the graph.

*Reply: Thank you for the correction. 97% of CE (Coulombic efficiency) should be referred to **Fig. 1D** not **Fig. 2B** and **2C**, the sentence has been modified as below in the revised main text. We also moved the label B away from enhancement part of the cell in **Fig. 2**.*

*“While the cell voltage deviated from the equilibrium potential (orange dash line in **Fig. 2B** and **2C**) as the current density increased, the discharge capacities did not vary (**Fig. 2B** and **2C**),*

which could be attributed to the enhanced reaction surface area endowed by wetting between carbon current collector and $\text{Cl}_2\text{-CCl}_4$ (Fig. 1B and 1C).”

30. Lines 127-131; compare the values of this battery with similar systems from the literature and highlight the advantages of the CFB system.

Reply: The performance matrices including energy density, energy efficiency, peak power, membrane cost and material cost for RFBs that published in recent 10 years are listed in Fig. 5C and Table S5 for through comparison. Therefore, we will not repeat them here.

31. Line 143; if the difference in the overpotentials for charge and discharge could only be attributed to mass transport effect, what is the cause?

Reply: The sentence has been modified to give detailed explanation on why the asymmetric overpotential is caused by mass transport (concentration gradient) in the revised main text:

“It is worth noting that polarizations for charge and discharge at the same current density are different (Fig. 2B and 2C), with larger overpotentials for discharge than those for charge. In the CFB, overpotentials are caused by redox reaction and concentration gradient. Since the symmetric factors for Cl^-/Cl_2 redox reactions are equal^{17,25}, the non-concentration induced overpotentials needed to drive the reduction and oxidation reaction are the same. The different overpotentials for charge and discharge observed here could only be attributed to concentration gradient.”

32. Line 160; please explain why the difference in chloride ion concentrations is larger than chlorine gas in the porous electrodes.

Reply: Simulation demonstrates a less efficient mass transport of Cl^- than Cl_2 in the porous carbon electrode. The potential reasons are : (1) the smaller diffusivity of Cl^- in water than that of Cl_2 in CCl_4 and (2) the lower volume percentage of the aqueous phase in the porous electrode than that of the CCl_4 phase. The explanation was highlighted in the revised main text:

“The maximum concentration differences of Cl^- are larger than those of Cl_2 in the porous $\text{RuO}_2\text{-TiO}_2\text{@C}$ electrode for both charge and discharge (Fig. 3C and 3D), which is the result of a smaller diffusivity of Cl^- ($1.5 \times 10^{-5} \text{ cm}^2/\text{s}$ for Cl^- , $2 \times 10^{-5} \text{ cm}^2/\text{s}$ for Cl_2 in NaCl electrolyte and $3 \times 10^{-5} \text{ cm}^2/\text{s}$ for Cl_2 in CCl_4 ⁵⁷⁻⁵⁹) and lower volume percentage of $\text{NaCl}/\text{H}_2\text{O}$ electrolyte than CCl_4 in the porous $\text{RuO}_2\text{-TiO}_2\text{@C}$ electrode.”

33. The axes of the inset curves in figures 3A and 3B should have labels or specify that they are the same as the main figure if they are.

Reply: The axis titles of insets in Fig. 3A and 3B are the same as the main figures, and they are added as shown in Fig. R4. Fig. 3 is replaced by Fig. R4 and highlighted in the revised main text to avoid confusion.

Figure R4. Simulation of the CFB. **(A)** Steady state potentials of CFB charged at 50% SOC with different Q_{aq} and $Q_{org} = 0.002$ mL/s, inset shows the whole current density range demonstrating steady charge potential. **(B)** Steady state potentials of CFB discharged at 50% SOC with different Q_{org} and $Q_{aq} = 0.02$ mL/s, inset shows the whole current density range demonstrating steady discharge potential. **(C)** Distribution of Cl^- and Cl_2 in the CFB charged at 50% SOC and 50 mA/cm² with $Q_{aq} = 0.02$ mL/s and $Q_{org} = 0.002$ mL/s. Upper legend indicates the positions of cathode, anode CCl_4 and aqueous phase in the CFB. **(D)** Distribution of Cl^- and Cl_2 in the CFB discharged at 50% SOC and 50 mA/cm² with $Q_{aq} = 0.02$ mL/s and $Q_{org} = 0.002$ mL/s. The position of Cl_2 - CCl_4 , $NaCl$ - H_2O , porous RuO_2 - TiO_2 @C positive electrode and $NaTi_2(PO_4)_3$ negative electrode are labeled in the legend.

34. Line 181; is there any way to provide data about how fast the kinetic rates are, even from the literature?

Reply: The reaction kinetics are characterized by activation energy E_a (kJ/mol) for reaction as demonstrated in question 8. The E_a for Cl/Cl_2 conversion in 4.0 M NaCl solution on RuO_2 is 35.5 kJ/mol (2). To provide a scale for comparison, the E_a for the extremely fast hydrogen evolution on Pt catalysis ranges between 9.5 to 18 kJ/mol (15).

35. Figures 3C, 3D, 4A and 4B should include labels to indicate the position of the anode and cathode.

*Reply: A legend has been added to **Fig. 3C** and **4A** to illustrate the position of cathode, anodes and the different liquid phases as shown in **Fig. R4** and **Fig. R5**. **Fig. 3** and **Fig. 4** are replaced by **Fig. R4** and **Fig. R5** respectively in the revised main text.*

Fig R5. The potential gradient in the electrolyte of CFB during (A) charge and (B) discharge at 50% SOC and 50 mA/cm^2 . (C) The potential loss due to ion transport in the electrolyte at different current densities. In all cases $Q_{aq} = 0.02$ mL/s and $Q_{org} = 0.002$ mL/s. The position of Cl_2-CCl_4 , $NaCl-H_2O$, porous $RuO_2-TiO_2@C$ positive electrode and $NaTi_2(PO_4)_3$ negative electrode are labeled in the legend.

36. Line 197; what is it included in the volume to calculate such value of the volumetric energy density?

*Reply: Upon careful examination, we found that the volume of the anode was not included in the calculation. Here we correct this error and provide the details on calculating the energy density of the CFB. In the 600 mAh cell used in this study, the average operating potential is 1.8 V at 10 mA/cm^2 , the volume of CCl_4 is 6 mL, the volume of $NaCl/H_2O$ electrolyte is 2 mL and the volume of $Na(Ti_2(PO_4)_3)$ is 0.592 mL (weight= 5g, density = 2.96 g/mL, volume = 2.96 g/mL \div 5g = 0.592 mL). The total volume of materials is 8.592 mL. Based on these configurations, the cell energy density (active material only) calculated with **equation R2** is 125.7 Wh/L.*

$$\text{Energy density} = \frac{\text{cell potential (V)} \times \text{capacity(Ah)}}{\text{Total volume of active materials (L)}} \quad (\text{R2})$$

The details for calculation were provided and highlighted in the revised Supplementary Materials as follow:

“ Energy density calculations

The energy density of CFB was calculated based on the 600 mAh cell used in this study, the average operating potential is 1.8 V at 10 mA/cm², the volume of CCl₄ is 6 mL, the volume of saturated NaCl solution is 2 mL and the volume of Na(Ti₂(PO₄)₃) is 0.592 mL (weight= 5g, density = 2.96 g/mL, volume = 2.96 g/mL ÷ 5g = 0.592 mL). The total volume of active materials is 8.592 mL . Based on these configurations, the cell level energy density (active material only) calculated with equation S1 is 125.7 Wh/L.

$$\text{Energy density} = \frac{\text{cell potential (V)} \times \text{capacity (Ah)}}{\text{Total volume of active materials (L)}} \quad (\text{S1})$$

And the value was called out and highlighted in the revised main text as follow:

“The CFB demonstrates the roundtrip energy efficiency of 91% (calculated by voltage efficiency x Coulombic efficiency) at 10 mA/cm² and provides an energy density of 125.7 Wh/L (see Methods in **Supplementary Materials**), which is among the highest of the flow battery systems reported in past 10 years (**Table S5**).”

37. Line 200. Include the word “the” before anode. Please revise through the document for similar missed articles, i.e. “the”.

Reply: The article “the” has been added throughout the revised manuscript.

Discussion section

38. Line 218-219; mention the other liquids that can replace CCl₄.

Reply: Other candidate solvents were stated and highlighted in the revised main text as follow:

“Other liquids that have high Cl₂ solubility and are phase-separated from NaCl/H₂O electrolytes can be used to replace CCl₄, including heptane (chlorine solubility = 0.173 mole fraction at ambient temperature), octane (chlorine solubility = 0.168 mole fraction at ambient temperature) and tetradecane (chlorine solubility = 0.254 mole fraction at ambient temperature)²⁹ and mineral spirit. Mineral spirit demonstrates good wettability (contact angle CA=9.1°) with carbon current collector (**Fig. S12A**), low viscosity (1.24 mPa.s), low toxicity and is cheaper than CCl₄⁶⁴. When CCl₄ was replaced by mineral spirit in the CFB, a volumetric capacity of 91.6 Ah/L was delivered at 20 °C (**Fig. S12B**).”

39. Line 223; correct the units after 20

Reply: The unit “°C” was added and highlighted in the revised main text.

40. Line 235; correct the error in the references after the words “free design”.

Reply: Thank you for pointing out the incorrect citation, the error in the references is corrected in the revised main text and highlighted as follow:

“Considering that the ion exchange membrane (mainly perfluorinated polymers) takes up more than 30% of the cost of flow batteries, significant cost reduction is expected with the membrane free design²⁰.”

41. Some references (36, 37) only mention the first author, et al. Is there any guideline for the numbers of authors or why not all the authors are named? Would not be fair to include all the authors that contributed to a certain reference?

Reply: Reference 36, 37 and 48 have more than five authors. According to the formatting guideline of Nature publication ((<https://www.nature.com/nature/for-authors/formatting-guide>), only the first author followed by et al. should be given in this case.

42. References to major revisions of RFB are missing for example:

<https://DOI.org/10.1021/bk-2020-1364.ch001>

<https://DOI.org/10.1039/c2ra21342g>

<https://doi.org/10.1016/j.jpowsour.2006.02.095>

Reply: The suggested references are added to and highlighted in the revised main text as reference 10, 7 and 5.

Reviewer #2 (Remarks to the Author):

Chen et al report a non-aqueous flow battery that does not require a membrane/separator due to the fact it has two immiscible (water and carbon tetrachloride) solvents. The battery also has a second significant advancement with facile molecular chlorine and chloride electrochemistry. This is a very elegant and thorough study. I am also quite impressed by the thoroughness of the supporting information. I have worked on non-aqueous flow battery research and development for more than ten years and I consider this to be in the top 10% most significant advancements. The major Achilles heel is my concern about the safety of using chlorine gas in a grid-scale (large-scale) battery. Safety is briefly mentioned right at the end of the manuscript, but I think it is a big mistake to just gloss over it, particularly with the broad readership of Nature Communications. A minor point is that the sentence structure (especially at the beginning of the manuscript) is awkward in places and could use some improvement.

Reply: Thank you very much for the positive comments. We have modified the discussion part and provided some potential safety administrations in the revised main text:

“Cl₂ is known to be reactive, but an important chemical commodity used in production of paper, plastics, dyes, textiles, medicines, antiseptics, insecticides, solvents, and paints. Administration and engineering controls for storage and transport are available to confine incident rate to 0.019 % of total chlorine shipments between 2007 and 2017⁶⁶. On the other hand, the Occupational Safety and Health Administration (OSHA) of the United States has set a permissible exposure limit (PEL) at a time-weighted average (TWA) of 0.1 ppm (0.68 mg/m³) for bromine, 0.05 mg/m³ for vanadium pentoxide dust, 0.1 ppm (0.4 mg/m³) for quinone, and 1 ppm (3 mg/m³) for chlorine⁶⁷, thus there are no obvious increase in chemical exposure risk when changing to chlorine redox reaction, although protection and caution are still important in container design. In this paper, the developed CFB is a closed system in which the leakage of Cl₂ gas can be well controlled with fluoropolymer gasket (see Supplementary Notes in **Supplementary Materials** for evaluation of chlorine permeation). In addition, strategies adapted from chloro-alkali industry can be applied to further reduce the risk of exposure upon scaling up, including external seal pipe, shutoff system, neutralizing reagents (scrubber)⁶⁸, sensing system to monitor the trace Cl₂ leakage and so forth⁶⁹.”

Reviewer #3 (Remarks to the Author):

Comments : In this paper, a new cell structure of membrane-free chlorine flow battery with high energy density and low-cost was developed. Both high solubility of Cl₂ in CCl₄ and immiscibility between CCl₄ and NaCl electrolyte enable (i) excellent flow battery cell performance with extremely high energy density, and (ii) membrane-free design endurable even at high current density. Additional mechanical modification that was carried out by adjusting flow rate could play a role in manipulating the unbalance of flow battery occurring by the difference in flowing speed accelerated by different phase of CCl₄ and NaCl. However, there are some points to be more clarified.

[1] In page 3, there is the sentence that “As the concentration of Cl₂ increases from zero to saturation, the viscosity even slightly decreases from 0.894 mPa.s to 0.819 mPa.s, while the viscosity of common catholyte could increase for several or even dozen times as the concentration of solute increases.” Authors should explain this result more specifically because they only mentioned the result itself. Especially, there is no explanation about how the viscosity of electrolytes became lower with increased concentration of solute such as Cl₂.

*Reply: Since both CCl₄ and Cl₂ are non-polar with little inter-molecular interactions, the Cl₂-CCl₄ mixture can be viewed as an ideal binary mixture, and the viscosity of Cl₂-CCl₄ mixture can be approximated with Eyring's absolute reaction rate theory, in which the logarithm of viscosity*volume is a weighted average of that from each component in the mixture as shown in equation R3 (16). Considering the Cl₂ gas has very small viscosity at room temperature (17), increasing the Cl₂ concentration (molar ratio) therefore reduces the overall viscosity of the Cl₂-CCl₄ mixture. Similar behavior was also observed by John R. Lewis that the viscosity of organic solvents slightly decreases when the SO₂ gas was mixed into them (18).*

$$\ln(\eta V) = x_1 \ln(\eta_1 V_1) + x_2 \ln(\eta_2 V_2) \quad (R3)$$

η = viscosity of the mixture, V = volume of the mixture, x_i = moles of species i in the mixture; η_i = viscosity of pure species i ; V_i = volume of pure species i

The explanation was added and highlighted in the revised manuscript:

“When the concentration of Cl₂ increases from 0 to 0.184 mole/mole CCl₄ (saturated concentration), the viscosity even slightly decreases from 0.894 mPa.s to 0.819 mPa.s (Fig. 1F), which is in accord to the Eyring's absolute reaction rate theory for gas-liquid mixture^{52,53}.”

[2] In page 5, there is the sentence that “Since Cl⁻ and Cl₂ are in different phases, the reaction kinetics of the cell can be further improved by increasing the flow rate of NaCl electrolyte during charge and that of the CCl₄ during discharge, ~.” The tested flow rate of NaCl electrolyte was 0.02, 0.1, 0.05 mL/s, and that of CCl₄ electrolyte was 0.0004, 0.002, 0.001 mL/s. Authors should explain why authors chose the range of flow rate. In addition, authors should explain the effect of optimal flow rate on the performances of flow battery cell because its cell voltage is sensitively affected by the flow rate.

Reply: The flow rate is chosen to ensure a stable interface for core-annular flow since high flow rate will cause a transition to slug flow in which aqueous phase will be segregated by CCl₄

phase in the flow direction (19). We have provided the discussion on choosing the flow rates in the revised manuscript and in the revised supporting materials:

“Since a continuous CCl_4 phase is needed to carry Cl_2 for redox reaction without interfering the ion transport in the aqueous phase, the flow rates chosen in this study are ones below the transition from core-annulus to a slug flow¹ in the cell.”

[3] In page 8, Fig. 5 shows the cell performance of flow battery cell during 50 cycles. However, this is too short because the $\text{NaTi}_2(\text{PO}_4)_3$ electrode takes advantage of long cycle life (1,000 cycles) as authors mentioned in this manuscript, Therefore, authors should include more long-cycling cell performance (>500 cycles) data to ensure the long lifetime of this flow battery cell system. As authors know, the long lifetime is important factor for flow battery cell system.

Reply: We agree with the reviewer’s concern. The cell was continued to run for 500 cycles, no significant loss of capacity was observed as in Fig. R6. Fig. 5B is replaced by Fig. R6 and highlighted in the revised main text.

Fig. R6. The capacity retention of the $\text{Cl}_2\text{-CCl}_4/\text{NaTi}_2(\text{PO}_4)_3$ flow cell under fully charged and discharged condition at 20 mA/cm^2

[4] In the introduction part, I recommend you add the paragraph explaining organic redox flow batteries as the example of one of promising RFB types with the following related references.

- (2020). Substituent pattern effects on the redox potentials of quinone-based active materials for aqueous redox flow batteries. *ChemSusChem*, 13(20), 5480-5488.
- (2020). Alkaline aqueous organic redox flow batteries of high energy and power densities using mixed naphthoquinone derivatives. *Chemical Engineering Journal*, 386, 123985.
- (2019). Extending the lifetime of organic flow batteries via redox state management. *Journal of the American Chemical Society*, 141(20), 8014-8019.
- (2020). Tailoring Dihydroxyphthalazines to Enable Their Stable and Efficient Use in the Catholyte of Aqueous Redox Flow Batteries. *Chemistry of Materials*, 32(8), 3427-3438.
- (2018). Alkaline quinone flow battery with long lifetime at pH 12. *Joule*, 2(9), 1894-1906.

Reply: Thank you for your comments, the references are added and highlighted in revised main text as reference 11-15.

Reference

1. Rahman, F., Skyllas-Kazacos, M., Solubility of vanadyl sulfate in concentrated sulfuric acid solutions. *J. Power Sources*. **2**, 105-110 (1998).
2. Janssen, L. J. J., Starmans, Li. M. C., Visser, J. G., Barendrecht, E. Mechanism of the chlorine evolution on a ruthenium oxide/titanium oxide electrode and on a ruthenium electrode. *Electrochimica Acta*, **22**, 1093-1100 (1977).
3. Agarwal, H., Florian, J., Goldsmith, B. R., Singh, N. V^{2+}/V^{3+} Redox Kinetics on Glassy Carbon in Acidic Electrolytes for Vanadium Redox Flow Batteries. *ACS Energy Lett.* **4**, 2368-2377 (2019).
4. Young, C. L. Sulfur dioxide. Chlorine, fluorine and chlorine oxides. *solubility data series*. **12**, 333-445 (1983).
5. Li, X., Jiang, C., Qian, Y., Liu, J., Yang, J., Xu, Q., Yan, C. Investigation of electrolytes of the vanadium redox flow battery (VII): Prediction of the viscosity of mixed electrolyte solution ($VOSO_4 + H_2SO_4 + H_2O$) based on Eyring's theory. *J. Chem. Thermodyn.* **134**, 69-75 (2019).
6. Dupont General Chemical Resistance Guide.
https://mscrm-dupont.secure.force.com/CRG_TlargaGuide
7. Sturm, P., Leuenberger, M., Sirignano, C., Neubert, R. E. M., Meijer, H. A. J., Langenfelds, R., Brand, W. A., Tohjima, Y. Permeation of atmospheric gases through polymer O-rings used in flasks for air sampling. *J. Geophys. Res. Atmos.* **109**, D04309 (2004).
8. Argazinski, J. K., Sant'Anna, J. A. P., Tristante, M. R. Fluoropolymers for the Chemical Processing Industry Applications. *3rd International Corrosion Meeting*, Fortaleza, Brazil (2010).
9. Occupational Safety and Health Administration (OSHA) set the permissible exposure limit data. Retrieved from <https://www.osha.gov/>
10. The Chlorine Manual. Sixth ed. Washington: The Chlorine Institute, INC., 2000.
11. Hildenbrand, D. L., McDonald, R. A., The Heat of Vaporization and Vapor Pressure of Carbon Tetrachloride; The Entropy from Calorimetric Data. *J. Phys. Chem.* **63**, 1521-1522 (1959).
12. Lyer, V. A. *et al.* Assessing the impact of electrolyte conductivity and viscosity on the reactor cost and pressure drop of redox-active polymer flow batteries. *J. Power Sources*. **361**, 334-344 (2017).
13. Sun, C., Chen, J., Zhang, H., Han, X., Luo, Q., Investigations on transfer of water and vanadium ions across Nafion membrane in an operating vanadium redox flow battery. *J. Power Sources*. **3**, 890-897 (2010).
14. Song, Y., Li, X., Yan, C., Tang, A. Unraveling the viscosity impact on volumetric transfer in redox flow batteries. *J. Power Sources*. **456**, 228004 (2020).

- 15 . Markovic, N. M., Grgur, B. N., Ross, P. N., Temperature-Dependent Hydrogen Electrochemistry on Platinum Low-Index Single-Crystal Surfaces in Acid Solutions. *J. Phys. Chem. B* , **101**, 5405–5413 (1997).
16. Bosse, D., Bart, H-J. Viscosity Calculations on the Basis of Eyring's Absolute Reaction Rate Theory and COSMOSPACE. *Ind. Eng. Chem. Res.* **44**, 8428-8435 (2005).
17. Engineering ToolBox. Gas-Dynamic Viscosity. https://www.engineeringtoolbox.com/gases-absolute-dynamic-viscosity-d_1888.html (2014)
18. Lewis, J. R. THE VISCOSITY OF LIQUIDS CONTAINING DISSOLVED GASES. *J. Am. Chem. Soc.* **47**, 626-640 (1925).
19. Brennen, C. E. *Fundamentals of Multiphase Flow*. 127-154 (Cambridge University Press, 2005).

REVIEWERS' COMMENTS

Reviewer #3 (Remarks to the Author):

I think that this revised version of the manuscript is ready to be published in Nat Com. The authors corrected well what I pointed out